# Deep Submodular Functions: Definitions & Learning

**Brian Dolhansky**[‡] <bdol@cs.washington.edu>      **Jeff Bilmes**[†‡] <bilmes@uw.edu>

Dept. of Computer Science and Engineering[‡]
University of Washington
Seattle, WA 98105

Dept. of Electrical Engineering[†]
University of Washington
Seattle, WA 98105

## Abstract

We propose and study a new class of submodular functions called *deep submodular functions* (DSFs). We define DSFs and situate them within the broader context of classes of submodular functions in relationship both to various matroid ranks and sums of concave composed with modular functions (SCMs). Notably, we find that DSFs constitute a strictly broader class than SCMs, thus motivating their use, but that they do not comprise all submodular functions. Interestingly, some DSFs can be seen as special cases of certain deep neural networks (DNNs), hence the name. Finally, we provide a method to learn DSFs in a max-margin framework, and offer preliminary results applying this both to synthetic and real-world data instances.

## 1 Introduction

Submodular functions are attractive models of many physical processes primarily because they possess an inherent naturalness to a wide variety of problems (e.g., they are good models of diversity, information, and cooperative costs) while at the same time they enjoy properties sufficient for efficient optimization. For example, submodular functions can be minimized without constraints in polynomial time [12] even though they lie within a $2^n$-dimensional cone in $\mathbb{R}^{2^n}$. Moreover, while submodular function maximization is NP-hard, submodular maximization is one of the easiest of the NP-hard problems since constant factor approximation algorithms are often available — e.g., in the cardinality constrained case, the classic $1 - 1/e$ result of Nemhauser [21] via the greedy algorithm. Other problems also have guarantees, such as submodular maximization subject to knapsack or multiple matroid constraints [8, 7, 18, 15, 16].

One of the critical problems associated with utilizing submodular functions in machine learning contexts is selecting which submodular function to use, and given that submodular functions lie in such a vast space with $2^n$ degrees of freedom, it is a non-trivial task to find one that works well, if not optimally. One approach is to attempt to learn the submodular function based on either queries of some form or based on data. This has led to results, mostly in the theory community, showing how learning submodularity can be harder or easier depending on how we judge what is being learnt. For example, it was shown that learning submodularity in the PMAC setting is fairly hard [2] although in some cases things are a bit easier [11]. In both of these cases, learning is over all points in the hypercube. Learning can be made easier if we restrict ourselves to learn within only a subfamily of submodular functions. For example, in [24, 19], it is shown that one can learn mixtures of submodular functions using a max-margin learning framework — here the components of the mixture are fixed and it is only the mixture parameters that are learnt, leading often to a convex optimization problem. In some cases, computing gradients of the convex problem can be done using submodular maximization [19], while in other cases, even a gradient requires minimizing a difference of two submodular functions [27].

Learning over restricted families rather than over the entire cone is desirable for the same reasons that any form of regularization in machine learning is useful. By restricting the family over which learning occurs, it decreases the complexity of the learning problem, thereby increasing the chance that one finds a good model within that family. This can be seen as a classic bias-variance tradeoff, where

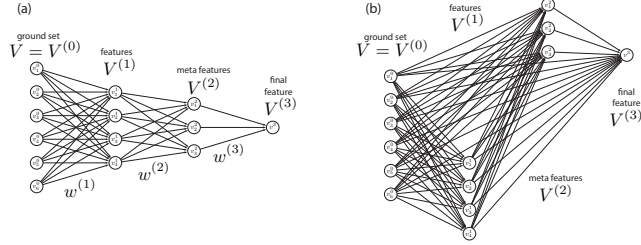

Figure 1: Left: A layered DSF with $K = 3$ layers. Right: a 3-block DSF allowing layer skipping.

increasing bias can reduce variance. Up to now, learning over restricted families has apparently (to the authors knowledge) been limited to learning mixtures over fixed components. This can be quite limited if the components are restricted, and if not might require a very large number of components. Therefore, there is a need for a richer and more flexible family of submodular functions over which learning is still possible.

In this paper (and in [5]), we introduce a new family of submodular functions that we term "deep submodular functions," or DSFs. DSFs strictly generalize, as we show below, many of the kinds of submodular functions that are useful in machine learning contexts. These include the so-called "decomposable" submodular functions, namely those that can be represented as a sum of concave composed with modular functions [25].

We describe the family of DSFs and place them in the context of the general submodular family. In particular, we show that DSFs strictly generalize standard decomposable functions, thus theoretically motivating the use of deeper networks as a family over which to learn. Moreover, DSFs can represent a variety of complex submodular functions such as laminar matroid rank functions. These matroid rank functions include the truncated matroid rank function [13] that is often used to show theoretical worst-case performance for many constrained submodular minimization problems. We also show, somewhat surprisingly, that like decomposable functions, DSFs are unable to represent all possible cycle matroid rank functions. This is interesting in and of itself since there are laminar matroids that are not cycle matroids. On the other hand, we show that the more general DSFs share a variety of useful properties with decomposable functions. Namely, that they: (1) can leverage the vast amount of practical work on feature engineering that occurs in the machine learning community and its applications; (2) can operate on multi-modal data if the data can be featurized in the same space; (3) allow for training and testing on distinct sets since we can learn a function from the feature representation level on up, similar to the work in [19]; and (4) are useful for streaming applications since functions can be evaluated without requiring knowledge of the entire ground set. These advantages are made apparent in Section 2.

Interestingly, DSFs also share certain properties with deep neural networks (DNNs), which have become widely popular in the machine learning community. For example, DNNs with weights that are strictly non-negative correspond to a DSF. This suggests, as we show in Section 5, that it is possible to develop a learning framework over DSFs leveraging DNN learning frameworks. Unlike standard deep neural networks, which typically are trained either in classification or regression frameworks, however, learning submodularity often takes the form of trying to adjust the parameters so that a set of "summary" data sets are offered a high value. We therefore extend the max-margin learning framework of [24, 19] to apply to DSFs. Our approach can be seen as a max-margin learning approach for DNNs but restricted to DSFs. We show that DSFs can be learnt effectively in a variety of contexts (Section 6). In the below, we discuss basic definitions and an initial implementation of learning DSFS while in [5] we provide complete definitions, properties, relationships to concavity, proofs, and a set of applications.

## 2   Background

Submodular functions are discrete set functions that have the *property of diminishing returns*. Assume a given finite size $n$ set of objects $V$ (the large *ground set* of data items), where each $v \in V$ is a distinct data sample (e.g., a sentence, training pair, image, video, or even a highly structured object such as a tree or a graph). A valuation set function $f : 2^V \to \mathbb{R}$ that returns a real value for any subset $X \subseteq V$ is said to be *submodular* if for all $X \subseteq Y$ and $v \notin Y$ the following inequality holds:

$f(X \cup \{v\}) - f(X) \geq f(Y \cup \{v\}) - f(Y)$. This means that the incremental value (or gain) of adding another sample $v$ to a subset decreases when the context in which $v$ is considered grows from $X$ to $Y$. We can define the *gain of $v$ in the context of $X$* as $f(v|X) \triangleq f(X \cup \{v\}) - f(X)$. Thus, $f$ is submodular if $f(v|X) \geq f(v|Y)$. If the gain of $v$ is identical for all different contexts i.e., $f(v|X) = f(v|Y), \forall X, Y \subseteq V$, then the function is said to be *modular*. A function might also have the property of being normalized ($f(\emptyset) = 0$) and monotone non-decreasing ($f(X) \leq f(Y)$ whenever $X \subseteq Y$). If the negation of $f$, $-f$, is submodular, then $f$ is called *supermodular*.

A useful class of submodular functions in machine learning are decomposable functions [25], and one example of useful instances of these for applications are called feature-based functions. Given a set of non-negative monotone non-decreasing normalized ($\phi(0) = 0$) concave functions $\phi_i : R_+ \to \mathbb{R}+$ and a corresponding set of non-negative modular functions $m_i : V \to \mathbb{R}_+$, the function $f : 2^V \to \mathbb{R}_+$ defined as $f(X) = \sum_i \phi_i(m_i(X)) = \sum_i \phi_i(\sum_{x \in X} m_i(x))$ is known to be submodular. Such functions have been called "decomposable" in the past, but in this work we will refer to them as the family of sums of concave over modular functions (SCMs). SCMs have been shown to be quite flexible [25], being able to represent a diverse set of functions such as graph cuts, set cover functions, and multiclass queuing system functions and yield efficient algorithms for minimization [25, 22]. Such functions are useful also for applications involving maximization. Suppose that each element $v \in V$ is associated with a set of "features" $U$ in the sense of, say, how TFIDF is used in natural language processing (NLP). Feature based submodular functions are those defined via the set of features. Feature based functions take the the form $f(X) = \sum_{u \in U} w_u \phi_u(m_u(X))$, where $\phi_u$ is a non-decreasing non-negative univariate normalized concave function, $m_u(X)$ is a feature-specific non-negative modular function, and $w_u$ is a non-negative feature weight. The result is the class of *feature-based submodular functions* (instances of SCMs). Such functions have been successfully used for data summarization [29].

Another advantage of such functions is that they do not require the construction of a pairwise graph and therefore do not have quadratic cost as would, say a facility location function (e.g., $f(X) = \sum_{v \in V} \max_{x \in X} w_{xv}$), or any function based on pair-wise distances, all of which have cost $O(n^2)$ to evaluate. Feature functions have an evaluation cost of $O(n|U|)$, linear in the ground set $V$ size and therefore are more scalable to large data set sizes. Finally, unlike the facility location and other graph-based functions, feature-based functions do not require the use of the entire ground set for each evaluation and hence are appropriate for streaming algorithms [1, 9] where future ground elements are unavailable. Defining $\psi : \mathbb{R}^n \to \mathbb{R}$ as $\psi(x) = \sum_i \phi_i(\langle m_i, x \rangle)$, we get a monotone non-decreasing concave function, which we refer to as univariate sum of concaves (USCs).

## 3  Deep Submodular Functions

While feature-based submodular functions are indisputably useful, their weakness lies in that features themselves may not interact, although one feature $u'$ might be partially redundant with another feature $u''$. For example, when describing a sentence via its component n-grams features, higher-order n-grams always include lower-order n-grams, so n-gram features are partially redundant. We may address this problem by utilizing an additional "layer" of nested concave functions as in $f(X) = \sum_{s \in S} \omega_s \phi_s(\sum_{u \in U} w_{s,u} \phi_u(m_u(X)))$, where $S$ is a set of meta-features, $\omega_s$ is a meta-feature weight, $\phi_s$ is a non-decreasing concave function associated with meta-feature $s$, and $w_{s,u}$ is now a meta-feature specific feature weight. With this construct, $\phi_s$ assigns a discounted value to the set of features in $U$, which can be used to represent feature redundancy. Interactions between the meta-features might be needed as well, and this can be done via meta-meta-features, and so on, resulting in a hierarchy of increasingly higher-level features.

We hence propose a new class of submodular functions that we call *deep submodular functions* (DSFs). They may make use of a series of disjoint sets (see Figure 1-(a)): $V = V^{(0)}$, which is the function's ground set, and additional sets $V^{(1)}, V^{(2)}, \ldots, V^{(K)}$. $U = V^{(1)}$ can be seen as a set of "features", $V^{(2)}$ as a set of meta-features, $V^{(3)}$ as a set of meta-meta features, etc. up to $V^{(K)}$. The size of $V^{(i)}$ is $d^i = |V^{(i)}|$. Two successive sets (or "layers") $i - 1$ and $i$ are connected by a matrix $w^{(i)} \in \mathbb{R}_+^{d^i \times d^{i-1}}$, for $i \in \{1, \ldots, K\}$. Given $v^i \in V^{(i)}$, define $w_{v^i}^{(i)}$ to be the row of $w^{(i)}$ corresponding to element $v^i$, and $w_{v^i}^{(i)}(v^{i-1})$ is the element of matrix $w^{(i)}$ at row $v^i$ and column $v^{i-1}$. We may think of $w_{v^i}^{(i)} : V^{(i-1)} \to \mathbb{R}_+$ as a modular function defined on set $V^{(i-1)}$. Thus, this matrix contains $d^i$ such modular functions. Further, let $\phi_{v^k} : \mathbb{R}_+ \to \mathbb{R}_+$ be a non-negative non-decreasing

concave function. Then, a $K$-layer DSF $f : 2^V \to \mathbb{R}_+$ can be expressed as follows, for any $A \subseteq V$:

$$f(A) = \phi_{vK}\left( \sum_{v^{K-1} \in V^{(K-1)}} w_{vK}^{(K)}(v^{K-1})\phi_{vK-1}\left( \cdots \sum_{v^2 \in V^{(2)}} w_{v3}^{(3)}(v^2)\phi_{v2}\left( \sum_{v^1 \in V^{(1)}} w_{v2}^{(2)}(v^1)\phi_{v1}\left( \sum_{a \in A} w_{v1}^{(1)}(a) \right) \right) \right) \right) \quad (1)$$

Submodularity follows since a composition of a monotone non-decreasing function $f$ and a monotone non-decreasing concave function $\phi$ $(g(\cdot) = \phi(f(\cdot)))$ is submodular (Theorem 1 in [20]) — a DSF is submodular via recursive application and since submodularity is closed under conic combinations.

A more general way to define a DSF (useful for the theorems below) uses recursion directly. We are given a directed acyclic graph (DAG) $\mathbf{G} = (\mathbf{V}, \mathbf{E})$ where for any given node $v \in \mathbf{V}$, we say $\mathrm{pa}(v) \subset \mathbf{V}$ are the parents of (vertices pointing towards) $v$. A given size $n$ subset of nodes $V \subset \mathbf{V}$ corresponds to the ground set and for any $v \in V$, $\mathrm{pa}(v) = \emptyset$. A particular "root" node $\imath \in \mathbf{V} \setminus V$ has the distinction that $\imath \notin \mathrm{pa}(q)$ for any $q \in \mathbf{V}$. Given a non-ground node $v \in \mathbf{V} \setminus V$, we define the concave function $\psi_v : \mathbb{R}^V \to \mathbb{R}_+$

$$\psi_v(x) = \phi_v\left( \sum_{u \in \mathrm{pa}(v) \setminus V} w_{uv}\psi_u(x) + \langle m_v, x \rangle \right) \quad (2)$$

where $\phi_v : \mathbb{R}_+ \to \mathbb{R}_+$ is a non-decreasing univariate concave function, $w_{uv} \in \mathbb{R}_+, m_v : \mathbb{R}^{\mathrm{pa}(v) \cap V} \to \mathbb{R}_+$ is a non-negative linear function that evaluates as $\langle m_v, x \rangle = \sum_{u \in \mathrm{pa}(v) \cap V} m_v(u)x(u))$ (i.e., $\langle m_v, x \rangle$ is a sparse dot-product over elements $\mathrm{pa}(v) \cap V \subseteq V$). The base case, where $\mathrm{pa}(v) \subseteq V$ therefore has $\psi_v(x) = \phi_v(\langle m_v, x \rangle)$, so $\psi_v(\mathbf{1}_A)$ is a SCM function with only one term in the sum ($\mathbf{1}_A$ is the characteristic vector of set $A$). A general DSF is defined as follows: for all $A \subseteq V$, $f(A) = \psi_\imath(\mathbf{1}_A) + m_\pm(A)$, where $m_\pm : V \to \mathbb{R}$ is an arbitrary modular function (i.e., it may include positive and negative elements). From the perspective of defining a submodular function, there is no loss of generality by adding the final modular function $m_\pm$ to a monotone non-decreasing function — this is because any submodular function can be expressed as a sum of a monotone non-decreasing submodular function and a modular function [10]. This form of DSF is more general than the layered approach mentioned above which, in the current form, would partition $\mathbf{V} = \{V^{(0)}, V^{(1)}, \ldots, V^{(K)}\}$ into layers, and where for any $v \in V^{(i)}$, $\mathrm{pa}(v) \subseteq V^{(i-1)}$. Figure 1-(a) corresponds to a layered graph $\mathbf{G} = (\mathbf{V}, \mathbf{E})$ where $r = v_1^3$ and $V = \{v_1^0, v_2^0, \ldots, v_6^0\}$. Figure 1-(b) uses the same partitioning but where units are allowed to skip by more than one layer at a time. More generally, we can order the vertices in $\mathbf{V}$ with order $\sigma$ so that $\{\sigma_1, \sigma_2, \ldots, \sigma_n\} = V$ where $n = |V|$, $\sigma_m = \imath = v^K$ where $m = |\mathbf{V}|$ and where $\sigma_i \in \mathrm{pa}(\sigma_j)$ iff $i < j$. This allows an arbitrary pattern of skipping while maintaining submodularity.

The layered definition in Equation (1) is reminiscent of feed-forward deep neural networks (DNNs) owing to its multi-layered architecture. Interestingly, if one restricts the weights of a DNN at every layer to be non-negative, then for many standard hidden-unit activation functions the DNN constitutes a submodular function when given Boolean input vectors. The result follows for any activation function that is monotone non-decreasing concave for non-negative reals, such as the sigmoid, the hyperbolic tangent, and the rectified linear functions. This suggests that DSFs can be trained in a fashion similar to DNNs, as is further developed in Section 5. The recursive definition of DSFs, in Equation (2) is more general and, moreover, useful for the analysis in Section 4.

DSFs should be useful for many applications in machine learning. First, they retain the advantages of feature-based functions (i.e., they require neither $O(n^2)$ nor access to the entire ground set for evaluation). Hence, DSFs can be both fast, and useful for streaming applications. Second, they allow for a nested hierachy of features, similar to advantages a deep model has over a shallow model. For example, a one-layer DSF must construct a valuation over a set of objects from a large number of low-level features which can lead to fewer opportunities for feature sharing while a deeper network fosters distributed representations, analogous to DNNs [3, 4]. Below, we show that DSFs constitute a strictly larger family of submodular functions than SCMs.

## 4 Analysis of the DSF family

DSFs represent a family that, at the very least, contain the family of SCMs. We argued intuitively that DSFs might extend SCMs as they allow components themselves to interact, and the interactions may propagate up a many-layered hierarchy. In this section, we formally place DSF within the context of more general submodular functions. We show that DSFs strictly generalize SCMs while preserving many of their attractive attributes. We summarize the results of this section in Figure 2, and that includes familial relationships amongst other classes of submodular functions (e.g., various matroid rank functions), useful for our main theorems.

Thanks to concave composition closure rules [6], the root function $\psi_{\wr}(x) : \mathbb{R}^n \to \mathbb{R}$ in Eqn. (2) is a monotone non-decreasing multivariate concave function that, by the concave-submodular composition rule [20], yields a submodular function $\psi_{\wr}(\mathbf{1}_A)$. It is widely known that **any** univariate concave function composed with non-negative modular functions yields a submodular function. However, given an arbitrary multivariate concave function this is not the case. Consider, for example, any concave function $\psi$ over $\mathbb{R}^2$ that offers the follow evaluations: $\psi(0,0) = \psi(1,1) = 1, \psi(0,1) = \psi(1,0) = 0$. Then $f(A) = \psi(\mathbf{1}_A)$ is not submodular. Given a multivariate concave function $\psi : \mathbb{R}^n \to \mathbb{R}$, the superdifferential $\partial\psi(x)$ at $x$ is defined as: $\partial\psi(x) = \{h \in \mathbb{R}^n : \psi(y) - \psi(x) \le \langle h, y \rangle - \langle h, x \rangle, \forall y \in \mathbb{R}^n\}$

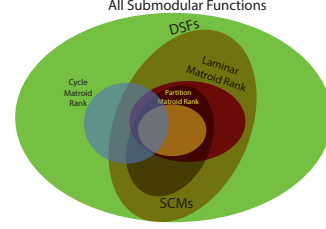

Figure 2: Containment properties of the set of functions studied in this paper.

and a particular supergradient $h_x$ is a member $h_x \in \partial\psi(x)$. If $\psi(x)$ is differentiable at $x$ then $\partial\psi(x) = \{\nabla\psi(x)\}$. A concave function is said to have an *antitone superdifferential* if for all $x \le y$ we have that $h_x \ge h_y$ for all $h_x \in \partial\psi(x)$ and $h_y \in \partial\psi(y)$ (here, $x \le y \Leftrightarrow x(v) \le y(v) \forall v$). Apparently, the following result has not been previously reported.

**Theorem 4.1.** *Let $\psi : \mathbb{R}^n \to \mathbb{R}$ be a concave function. Then if $\psi$ has an antitone superdifferential, then the set function $f : 2^V \to \mathbb{R}$ defined as $f(A) = \psi(\mathbf{1}_A)$ for all $A \subseteq V$ is submodular.*

A DSF's associated concave function has an antitone superdifferential. Concavity is not necessary in general (e.g., multilinear extensions of submodular functions are not concave but have properties analogous to an antitone superdifferential, see [5]).

**Lemma 4.3.** *Composition of monotone non-decreasing scalar concave and antitone superdifferential concave functions, and conic combinations thereof, preserves superdifferential antitonicity.*

**Corollary 4.3.1.** *The concave function $\psi_{\wr}$ associated with a DSF has an antitone superdifferential.*

A matroid $M$ [12] is a set system $M = (V, \mathcal{I})$ where $\mathcal{I} = \{I_1, I_2, \ldots\}$ is a set of subsets $I_i \subseteq V$ that are called independent. A matroid has the property that $\emptyset \in \mathcal{I}$, that $\mathcal{I}$ is subclusive (i.e., given $I \in \mathcal{I}$ and $I' \subset I$ then $I' \in \mathcal{I}$) and that all maximally independent sets have the same size (i.e., given $A, B \in \mathcal{I}$ with $|A| < |B|$, there exists a $b \in B \setminus A$ such that $A + b \in \mathcal{I}$). The rank of a matroid, a set function $r : 2^V \to \mathbb{Z}_+$ defined as $r(A) = \max_{I \in \mathcal{I}} |I \cap A|$, is a powerful class of submodular functions. All matroids are uniquely defined by their rank function. All monotone non-decreasing non-negative rational submodular functions can be represented by grouping and then evaluating grouped ground elements in a matroid [12].

A particularly useful matroid is the partition matroid, where a partition $(V_1, V_2, \ldots, V_\ell)$ of $V$ is formed, along with a set of capacities $k_1, k_2, \ldots, k_\ell \in \mathbb{Z}_+$. It's rank function is defined as: $r(X) = \sum_{i=1}^{\ell} \min(|X \cap V_i|, k_i)$ and, therefore, is an SCM, owing to the fact that $\phi(x) = \min(\langle x, \mathbf{1}_{V_i} \rangle, k_i)$ is USC. A cycle matroid is a different type of matroid based on a graph $G = (V, E)$ where the rank function $r(A)$ for $A \subseteq E$ is defined as the size of the maximum spanning forest (i.e., a spanning tree for each connected component) in the edge-induced subgraph $G_A = (V, A)$. From the perspective of matroids, we can consider classes of submodular functions via their rank. If a given type of matroid cannot represent another kind, their ranks lie in distinct families. To study where DSFs are situated in the space of all submodular functions, it is useful first to study results regarding matroid rank functions.

**Lemma 4.4.** *There are partition matroids that are not cycle matroids.*

In a *laminar matroid*, a generalization of a partition matroid, we start with a set $V$ and a family $\mathcal{F} = \{F_1, F_2, \ldots, \}$ of subsets $F_i \subseteq V$ that is *laminar*, namely that for all $i \ne j$ either $F_i \cap F_j = \emptyset$ or $F_i \subseteq F_j$ or $F_j \subseteq F_i$ (i.e., sets in $\mathcal{F}$ are either non-intersecting or comparable). In a laminar matroid, we also have for every $F \in \mathcal{F}$ an associated capacity $k_F \in \mathbb{Z}_+$. A set $I$ is independent if $|I \cap F| \le k_F$ for all $F \in \mathcal{F}$. A laminar family of sets can be organized in a tree, where there is one root $R \in \mathcal{F}$ in the tree that, w.l.o.g., can be $V$ itself. Then the immediate parents $\text{pa}(F) \subset \mathcal{F}$ of a set $F \in \mathcal{F}$ in the tree are the set of maximal subsets of $F$ in $\mathcal{F}$, i.e., $\text{pa}(F) = \{F' \in \mathcal{F} : F' \subset F \text{ and } \nexists F'' \in \mathcal{F} \text{ s.t. } F' \subset F'' \subset F\}$. We then define the following for all $F \in \mathcal{F}$:

$$r_F(A) = \min\left( \sum_{F' \in \text{pa}(F)} r_{F'}(A \cap F') + |A \setminus \bigcup_{F' \in \text{pa}(F)} F'|, k_F \right). \tag{3}$$

A laminar matroid rank has a recursive definition $r(A) = r_R(A) = r_V(A)$. Hence, if the family $\mathcal{F}$ forms a partition of $V$, we have a partition matroid. More interestingly, when compared to Eqn. (2), we see that a laminar matroid rank function is an instance of a DSF with a tree-structured DAG rather

than the non-tree DAGs in Figure 1. Thus, within the family of DSFs lie the truncated matroid rank functions used to show information theoretic hardness for many constrained submodular optimization problems [13]. Moreover, laminar matroids strictly generalize partition matroids.

**Lemma 4.5.** *Laminar matroids strictly generalize partition matroids*

Since a laminar matroid generalizes a partition matroid, this portends well for DSFs generalizing SCMs. Before considering that, we already are up against some limits of laminar matroids, i.e.:

**Lemma 4.6** (peeling proof). *Laminar matroid cannot represent all cycle matroids.*

We call this proof a "peeling proof" since it recursively peels off each layer (in the sense of a DSF) of a laminar matroid rank until it boils down to a partition matroid rank function, where the base case is clear. The proof is elucidating, moreover, since it motivates the proof of Theorem 4.14 showing that DSFs extend SCMs. We also have the immediate corollary.

**Corollary 4.6.1.** *Partition matroids cannot represent all cycle matroids.*

We see that SCMs generalize partition matroid rank functions and DSFs generalize laminar matroid rank functions. We might expect, from the above results, that DSFs might generalize SCMs — this is not immediately obvious since SCMs are significantly more flexible than partition matroid rank functions because: (1) the concave functions need not be simple truncations at integers, (2) each term can have its own non-negative modular function, (3) there is no requirement to partition the ground elements over terms in an SCM, and (4) we may with relative impunity extend the family of SCMs to ones where we add an additional arbitrary modular function (what we will call SCMMs below). We see, however, that SCMMs are also unable to represent the cycle matroid rank function over $K_4$, very much like the partition matroid rank function. Hence the above flexibility does not help in this case. We then show that DSFs strictly generalize SCMMs, which means that DSFs indeed provide a richer family of submodular functions to utilize, ones that as discussed above, retain many of the advantages of SCMMs. We end the section by showing that DSFs, even with an additional arbitrary modular function, are still unable to represent matroid rank over $K_4$, implying that although DSFs extend SCMMs, they cannot express all monotone non-decreasing submodular functions.

We define a family of sums of concave over modular functions with an additional modular term (SCMMs), taking the form: $f(A) = \sum_i \phi_i(m_i(A)) + m_\pm(A)$ where each $\phi_i$ and $m_i$ as in an SCM, but where $m_\pm : 2^V \to \mathbb{R}$ is an arbitrary modular function, so if $m_\pm(\cdot) = 0$ the SCMM is an SCM.

Before showing that DSFs extend SCMMs, we include a result showing that SCMMs are strictly smaller than the set of all submodular functions. We include an unpublished result [28] showing that SCMMs can not represent the cycle matroid rank function, as described above, over the graph $K_4$.

**Theorem 4.11** (Vondrak[28]). *SCMMs $\subset$ Space of Submodular Functions*

We next show that DSFs strictly generalize SCMMs, thus providing justification for using DSFs over SCMMs and, moreover, generalizing Lemma 4.5. The DSF we choose is, again, a laminar matroid, so SCMMs are unable to represent laminar matroid rank functions. Since DSFs generalize laminar matroid rank functions, the result follows.

**Theorem 4.14.** *The DSF family is strictly larger than that of SCMs.*

The proof shows that it is not possible to represent a function of the form $f(X) = \min(\sum_i \min(|X \cap B_i|, k_i), k)$ using an SCMM. Theorem 4.15 also has an immediate consequence for concave functions.

**Corollary 4.14.1.** *There exists a non-USC concave function with an antitone superdifferential.*

The corollary follows since, as mentioned above, DSF functions have at their core a multivarate concave function with an antitone superdifferential, and thanks to Theorem 4.14 it is not always possible to represent this as a sum of concave over linear functions. It is currently an open problem if DSFs with $\ell$ layers extend the family of DSFs with $\ell' < \ell$ layers, for $\ell' \geq 2$. Our final result shows that, while DSFs are richer than SCMMs, they still do not encompass all polymatroid functions. We show this by proving that the cycle matroid rank function on $K_4$ is not achievable with DSFs.

**Theorem 4.15.** *DSFs $\subset$ Polymatroids*

Proofs of these theorems and more may be found in [5].

# 5 Learning DSFs

As mentioned above, learning submodular functions is generally difficult [11, 13]. Learning mixtures of fixed submodular component functions [24, 19], however, can give good empirical results on several tasks, including image [27] and document [19] summarization. In these examples, rather than attempting to learn a function at all $2^n$ points, a max-margin approach is used only to approximate a submodular function on its large values. Typically when training a summarizer, one is given a ground set of items, and a set of representative sets of excerpts (usually human generated) each of which summarizes the ground set. Within this setting, access to an oracle function $h(A)$ — that, if available, could be used in a regression-style learning approach — might not be available. Even if available, such learning is often overkill. Thus, instead of trying to learn $h$ everywhere, we only seek to learn the parameters $\mathbf{w}$ of a function $f_{\mathbf{w}}$ that lives within some family, parameterized by $\mathbf{w}$, so that if $B \in \mathrm{argmax}_{A \subseteq V : |A| \leq k} f_{\mathbf{w}}(A)$, then $h(B) \geq \alpha h(A^*)$ for some $\alpha \in [0, 1]$ where $A^* \in \mathrm{argmax}_{A \subseteq V : |A| \leq k} h(A)$. In practice, this corresponds to selecting the best summary for a given document based on the learnt function, in the hope that it mirrors what a human believes to be best. Fortunately, the max-margin training approach directly addresses the above and is immediately applicable to learning DSFs. Also, given the ongoing research on learning DNNs, which have achieved state-of-the-art results on a plethora of machine learning tasks [17], and given the similarity between DSFs and DNNs, we may leverage the DNN learning techniques (such as dropout, AdaGrad, learning rate scheduling, etc.) to our benefit.

## 5.1 Using max-margin learning for DSFs

Given an unknown but desired function $h : 2^V \rightarrow \mathbb{R}_+$ and a set of representative sets $\mathcal{S} = \{S_1, S_2, \ldots\}$, with $S_i \subseteq V$ and where for each $S \in \mathcal{S}$, $h(S)$ is highly scored, a max-margin learning approach may be used to train a DSF $f$ so that if $A \in \mathrm{argmax}_{A \subseteq V} f(A)$, $h(A)$ is also highly scored by $h$. Under the large-margin approach [24, 19, 27], we learn the parameters $\mathbf{w}$ of $f_{\mathbf{w}}$ such that for all $S \in \mathcal{S}$, $f_{\mathbf{w}}(S)$ is high, while for $A \in 2^V$, $f_{\mathbf{w}}(A)$ is lower by some given loss. This may be performed by maximizing the loss-dependent margin so that for all $S \in \mathcal{S}$ and $A \in 2^V$, $f_{\mathbf{w}}(S) \geq f_{\mathbf{w}}(A) + \ell_S(A)$. For a given loss function $\ell_S(A)$, optimization reduces to finding parameters so that $f_{\mathbf{w}}(S) \geq \max_{A \in 2^V} [f_{\mathbf{w}}(A) + \ell_S(A)]$ is satisfied for $S \in \mathcal{S}$. The task of finding the maximizing set is known as loss-augmented inference (LAI) [26], which for general $\ell(A)$ is NP-hard. With regularization, and defining the hinge operator $(x)^+ = \max(0, x)$, the optimization becomes:

$$\min_{\mathbf{w} \geq 0} \sum_{S \in \mathcal{S}} \left( \max_{A \in 2^V} [f(A) + \ell_S(A)] - f(S) \right)^+ + \frac{\lambda}{2} ||\mathbf{w}||_2^2. \tag{4}$$

Given a LAI procedure, the subgradient of weight $w_i$ is $\frac{\partial}{\partial w_i} f(A) - \frac{\partial}{\partial w_i} f(S) + \lambda w_i$, and in the case of a DSF, each subgradient can be computed efficiently with backpropagation, similar to the approach of [23], but to retain polymatroidality of $f$, projected gradient descent is used ensure $\mathbf{w} \succeq 0$

For arbitrary set functions, $f(A)$ and $\ell_S(A)$, LAI is generally intractable. Even if $f(A)$ is submodular, the choice of loss can affect the computational feasibility of LAI. For submodular $\ell(A)$, the greedy algorithm can find an approximately maximizing set [21]. For supermodular $\ell(A)$, the task of solving $\max_{A \in 2^V \setminus \mathcal{S}} [f(A) + \ell(A)]$ involves maximizing the difference of two submodular functions and the submodular-supermodular procedure [14] can be used.

Once a DSF is learnt, we may wish to find $\max_{A \subseteq V : |A| \leq k} f(A)$ and this can be done, e.g., using the greedy algorithm when $m_{\pm} \geq 0$. The task of summarization, however, might involve learning based on one set of ground sets and testing via a different (set of) ground set(s) [19, 27]. To do this, any particular element $v \in V$ may be represented by a vector of non-negative feature weights $(m_1(v), m_2(v), \ldots)$ (e.g., $m_i(v)$ counts the number of times a unigram $i$ appears in sentence $v$), and the feature $i$ weight for any set $A \subseteq V$ can is represented as the $i$-specific modular evaluation $m_i(A) = \sum_{a \in A} m_i(a)$. We can treat the set of modular functions $\{m_i : V \rightarrow \mathbb{R}_+\}_i$ as a matrix to be used as the first layer in DSF (e.g., $w^{(1)}$ in Figure 1 (left)) that is fixed during the training of subsequent layers. This preserves submodularity, and allows all later layers (i.e., $w^{(2)}, w^{(3)}, \ldots$) to be learnt generically over any set of objects that can be represented in the same feature space — this also allows training over one set of ground sets, and testing on a totally separate set of ground sets. Max-margin learning, in fact, remains ignorant that this is happening since it sees the data only post feature representation. In fact, learning can be cross modal — e.g., images and sentences, represented in the same feature space, can be learnt simultaneously. This is analogous to the "shells" of [19]. In

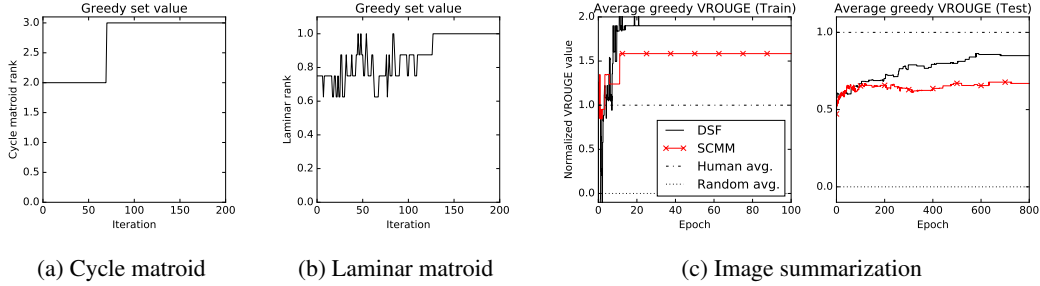

| (a) Cycle matroid | (b) Laminar matroid | (c) Image summarization |

Figure 3: (a),(b) show matroid learning via a DSF is possible in a max-margin setting; (c) shows that learning and generalization of a DSF can happen, via featurization, on real image data.

that case, however, mixtures were learnt over fixed components, some of which required a $O(n^2)$ calculation for element-pair similarity scores. Via featurization in the first layer of a DSF, however, we may learn a DSF over a training set, preserving submodularity, avoid any $O(n^2)$ cost, and test on any new data represented in the same feature space.

## 6 Empirical Experiments on Learning DSFs

We offer preliminary feasibility results showing it is possible to train a DSF on synthetic datasets and, via featurization, on a real image summarization dataset.

The first synthetic experiment trains a DSF to learn a cycle matroid rank function on $K_4$. Although Theorem 4.15 shows a DSF cannot represent such a rank function everywhere, we show that the max-margin framework can learn a DSF that, when maximized via $\max_{A \subseteq V:|A|\leq 3} f(A)$ does not return a 3-cycle (as is desirable). We used a simple two-layer DSF, where the first hidden layer consisted of four hidden units with square root activation functions, and a normalized sigmoid $\hat{\sigma}(x) = 2 \cdot (\sigma(x) - 0.5)$ at the output. Figure 3a shows that after sufficient learning iterations, greedy applied to the DSF returns independent sized-three sets. Further analysis shows that the function is not learnt everywhere, as predicted by Theorem 4.15.

We next tested a scaled laminar matroid rank $r(A) = (1/8) \min\left(\sum_{i=1}^{10} \min(|A \cap B_i|, 1), 8\right)$ where the $B_i$'s are each size 10 and form a partition of $V$, with $|V| = 100$. Thus maximal independent sets $\operatorname{argmax}_{I \in \mathcal{I}} |I|$ have $r(I) = 1$ with $|I| = 8$. A DSF is trained with a hidden layer of 10 units of activation $g(x) = \max(x, 1)$, and a normalized sigmoid $\hat{\sigma}$ at the output. We randomly generated 200 matroid bases, and trained the network. The greedy solution to $\max_{A \subseteq V:|A|\leq 8} f(A)$ on the learnt DSF produces sets that are maximally independent (Figure 3b).

For our real-world instance of learning DSFs, we use the dataset of [27], which consists of 14 distinct image sets, 100 images each. The task is to select the best 10-image summary in terms of a visual ROUGE-like function that is defined over a bag of visual features. For each of the 14 ground sets, we trained on the other 13 sets and evaluated the performance of the trained DSF on the test set. We use a simple DSF of the form $f(A) = \hat{\sigma}\left(\sum_{u \in U} w_u \sqrt{m_u(A)}\right)$, where $m_u(A)$ is modular for feature $u$, and $\hat{\sigma}$ is a sigmoid. We used (diagonalized) Adagrad, a decaying learning rate, weight decay, and dropout (which was critical for test-set performance). We compared to an SCMM of comparable complexity and number of parameters (i.e., the same form and features but a linear output), and performance of the SCMM is much worse (Figure 3c) perhaps because of a DSF's "depth." Notably, we only require $|U| = 628$ visual-word features (as covered in Section 5 of [27]), while the approach in [27] required 594 components of $O(n^2)$ graph values, or roughly 5.94 million precomputed values. The loss function is $\ell(A) = 1 - \mathcal{R}(A)$, where $\mathcal{R}(A)$ is a ROUGE-like function defined over visual-words. During training, we achieve numbers comparable to [27]. We do not yet match the generalization results in [27], but we do not use strong $O(n^2)$ graph components, and we expect better results perhaps with a deeper network and/or better base features.

Acknowledgments: Thanks to Reza Eghbali and Kai Wei for useful discussions. This material is based upon work supported by the National Science Foundation under Grant No. IIS-1162606, the National Institutes of Health under award R01GM103544, and by a Google, a Microsoft, a Facebook, and an Intel research award. This work was supported in part by TerraSwarm, one of six centers of STARnet, a Semiconductor Research Corporation program sponsored by MARCO and DARPA.

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
