[Reviews · NeurIPS 2016]

Reviewer 1

Summary

Problem definition - The paper proposes a new family of submodular functions called deep submodular functions. (DSF). They are defined similar to a neural network where there are many nodes in each level. At each node you take a positive linear combination of previous layer and then apply a concave function. Contributions - The main important contribution of the paper is proposing the family of DSF and showing applications to text summarization. - They show that DSF strictly generalize SCM (concave composed with modular functions) which have been well studied before. - They characterize DSF's in the hierarchy of other class of submodular functions such as partition matroid rank, laminar matroid rank, cycle matroid rank and SCMM. They show that DSF's generalize all of them except cycle matroid rank. - The paper shows how previous approaches to learning SCM's using max margin approach can be used to learn DSF and also use other training techniques for learning deep neural networks to learn DSF. They experimentally show how this can lead to better results for summarization over SCMM.

Qualitative Assessment

A nice paper. To make the paper even better the authors can do the following two. a) Evaluate the results on a much much larger data set. The power of deep functions is more evident in such situations. (as noted in the conclusions). b) They can show examples of summaries for both SCM's and DSF's to show the difference. (at least in supplementary section).

Confidence in this Review

2-Confident (read it all; understood it all reasonably well)


Reviewer 2

Summary

This paper proposes a new class of submodular function, which can be represented by multiple-layers of nest concave functions. It is named "Deep submodular functions (DSFs)". The authors discuss the properties of DSFs and especially they showed DSFs subsume the sum of concave functions. Also, the similarity between DSFs framework and Deep Neural Networks(DNNs) is discussed. Beyond analysis of the proposed class of functions, a numerical framework to learn DSFs is presented using max-margin learning. Also, as a proof of concept, experiments are conducted on synthetic data and real data set.

Qualitative Assessment

This paper proposes a new class of submodular functions. Also it is more general than decomposable functions or sums of concave over modular functions (SCMs). This result seems quite intuitive since SCMs is a special case of DSFs as a single layer NN is a special case of DNNs. However, the concave composition closure rules cast a question whether DSFs is strictly larger than SCMs. Thm 4.14 in the paper, clearly shows that DSFs are more flexible and worth to study. This is an exciting result and will improve the applicability of submodular frameworks. The only weakness of this paper is empirical performance in real applications as authors admit. But this is understandable for theoretical/analysis papers. It will be very interesting to see how this result impacts deep learning/submodular researches and applications of this result.

Confidence in this Review

2-Confident (read it all; understood it all reasonably well)


Reviewer 3

Summary

On the one hand, this paper proposed a certain class of submodular functions called deep submodular functions (DSF), with thorough analysis of the generalization property of DSF compared with other matroid ranks. On the other hand, the DSF owns DNN-like structure and could be trained in a max-margin framework using back-propagation. Experiments verified the findings and properties.

Qualitative Assessment

The paper organized very well, including a thorough introduction and analysis of deep submodular function, and a learning framework that exploits its own structural advantage. It would be more encouraging if state-of-art application results could be achieved with this framework.

Confidence in this Review

2-Confident (read it all; understood it all reasonably well)


Reviewer 4

Summary

The papers introduces deep submodular functions which is a generalization of previous considered sums of concave over modular functions (SCM). SCM's are basically submodular functions obtained by summing over concave functions applied to the output of a sum of modular/linear functions. Now the deep submodular functionsare sum of concave sum of concave ... on sum of modular functions. This seems like a natural generalization of SCMs. The paper proves that deep submodular functions is a strictly bigger class than SCMs and a strict subset of submodular functions. They also argue that these functions are easier to optimize than submodular functions in general. In particular, they consider the problem of learning a deep submodular function. These results are experimental (and no real proofs).

Qualitative Assessment

The novelty in terms of techniques seems limited and the theory results not too surprising. I would say that the main contribution of the paper is the introduction of deep submodular functions. This is definitely a worthwhile conceptual contribution. But, as the paper contains limited indications of its usefulness, I rate this contribution as insufficient for NIPS. I would also recommend that the authors are more careful in some of the writing. For example, in the first paragraph you say that submodular function maximization is one of the easiest NP-hard problems and then cite the result of Nemhauser's (1-1/e)-approximation. The best approximation for submodular function maximization is 1/2 and that is tight. Nemhauser result is in the monotone case and then it is trivial to maximize; his result is when you have a cardinality constraint. Also, I would say that the rank function of the laminar matroid is a fairly simple submodular function.

Confidence in this Review

2-Confident (read it all; understood it all reasonably well)